# Learning to Share and Hide Intentions using Information Regularization

**DJ Strouse**[1], **Max Kleiman-Weiner**[2], **Josh Tenenbaum**[2]
**Matt Botvinick**[3,4], **David Schwab**[5]
[1] Princeton University, [2] MIT, [3] DeepMind
[4] UCL, [5] CUNY Graduate Center

## Abstract

Learning to cooperate with friends and compete with foes is a key component of multi-agent reinforcement learning. Typically to do so, one requires access to either a model of or interaction with the other agent(s). Here we show how to learn effective strategies for cooperation and competition in an asymmetric information game with no such model or interaction. Our approach is to encourage an agent to reveal or hide their intentions using an information-theoretic regularizer. We consider both the mutual information between goal and action given state, as well as the mutual information between goal and state. We show how to optimize these regularizers in a way that is easy to integrate with policy gradient reinforcement learning. Finally, we demonstrate that cooperative (competitive) policies learned with our approach lead to more (less) reward for a second agent in two simple asymmetric information games.

## 1 Introduction

In order to effectively interact with others, an intelligent agent must understand the intentions of others. In order to successfully cooperate, collaborative agents that share their intentions will do a better job of coordinating their plans together [Tomasello et al., 2005]. This is especially salient when information pertinent to a goal is known asymmetrically between agents. When competing with others, a sophisticated agent might aim to hide this information from its adversary in order to deceive or surprise them. This type of sophisticated planning is thought to be a distinctive aspect of human intelligence compared to other animal species [Tomasello et al., 2005].

Furthermore, agents that share their intentions might have behavior that is more interpretable and understandable by people. Many reinforcement learning (RL) systems often plan in ways that can seem opaque to an observer. In particular, when an agent's reward function is not aligned with the designer's goal the intended behavior often deviates from what is expected [Hadfield-Menell et al., 2016]. If these agents are also trained to share high-level and often abstract information about its behavior (i.e. intentions) it is more likely a human operator or collaborator can understand, predict, and explain that agents decision. This is key requirement for building machines that people can trust.

Previous approaches have tackled aspects of this problem but all share a similar structure [Dragan et al., 2013, Ho et al., 2016, Hadfield-Menell et al., 2016, Shafto et al., 2014]. They optimize their behavior against a known model of an observer which has a theory-of-mind [Baker et al., 2009, Ullman et al., 2009, Rabinowitz et al., 2018] or is doing some form of inverse-RL [Ng et al., 2000, Abbeel and Ng, 2004]. In this work we take an alternative approach based on an information theoretic formulation of the problem of sharing and hiding intentions. This approach does not require an explicit model of or interaction with the other agent, which could be especially useful in settings where interactive training is expensive or dangerous. Our approach also naturally combines with scalable policy-gradient methods commonly used in deep reinforcement learning.

## 2  Hiding and revealing intentions via information-theoretic regularization

We consider multi-goal environments in the form of a discrete-time finite-horizon discounted Markov decision process (MDP) defined by the tuple $\mathcal{M} \equiv (\mathcal{S}, \mathcal{A}, \mathcal{G}, P, \rho_G, \rho_S, r, \gamma, T)$, where $\mathcal{S}$ is a state set, $\mathcal{A}$ an action set, $P : \mathcal{S} \times \mathcal{A} \times \mathcal{S} \to \mathbb{R}_+$ a (goal-independent) probability distribution over transitions, $\mathcal{G}$ a goal set, $\rho_G : \mathcal{G} \to \mathbb{R}_+$ a distribution over goals, $\rho_S : \mathcal{S} \to \mathbb{R}_+$ a probability distribution over initial states, $r : \mathcal{S} \times \mathcal{G} \to \mathbb{R}$ a (goal-dependent) reward function $\gamma \in [0, 1]$ a discount factor, and $T$ the horizon.

In each episode, a goal is sampled and determines the reward structure for that episode. One agent, Alice, will have access to this goal and thus knowledge of the environment's reward structure, while a second agent, Bob, will not and instead must infer it from observing Alice. We assume that Alice knows in advance whether Bob is a friend or foe and wants to make his task easier or harder, respectively, but that she has no model of him and must train without any interaction with him.

Of course, Alice also wishes to maximize her own expected reward $\eta[\pi] = \mathbb{E}_\tau \left[ \sum_{t=0}^{T} \gamma^t r(s_t, g) \right]$, where $\tau = (g, s_0, a_0, s_1, a_1, \ldots, s_T)$ denotes the episode trajectory, $g \sim \rho_G$, $s_0 \sim \rho_S$, $a_t \sim \pi_g(a_t \mid s_t)$, and $s_{t+1} \sim P(s_{t+1} \mid s_t, a_t)$, and $\pi_g(a \mid s; \theta) : \mathcal{G} \times \mathcal{S} \times \mathcal{A} \to \mathbb{R}_+$ is Alice's goal-dependent probability distribution over actions (policy) parameterized by $\theta$.

It is common in RL to consider loss functions of the form $J[\pi] = \eta[\pi] + \beta \ell[\pi]$, where $\ell$ is a regularizer meant to help guide the agent toward desirable solutions. For example, the policy entropy is a common choice to encourage exploration [Mnih et al., 2016], while pixel prediction and control have been proposed to encourage exploration in visually rich environments with sparse rewards [Jaderberg et al., 2017].

The setting we imagine is one in which we would *like* Alice to perform well in a joint environment with rewards $r_{\text{joint}}$, but we are only able to train her in a solo setting with rewards $r_{\text{solo}}$. How do we make sure that Alice's learned behavior in the solo environment transfers well to the joint environment? We propose the training objective $J_{\text{train}} = \mathbb{E}[r_{\text{solo}}] + \beta I$ (where I is some sort of task-relevant information measure) as a useful for proxy for the test objective $J_{\text{test}} = \mathbb{E}[r_{\text{joint}}]$. The structure of $r_{\text{joint}}$ determines whether the task is cooperative or competitive, and therefore the appropriate sign of $\beta$. For example, in the spatial navigation game of section 4.1, a competitive $r_{\text{joint}}$ might provide +1 reward *only* to the first agent to reach the correct goal (and -1 for reaching the wrong one), whereas a cooperative $r_{\text{joint}}$ might provide each of Alice and Bob with the sum of their individual rewards. In figure 2, we plot related metrics, after training Alice with $J_{\text{train}}$. On the bottom row, we plot the percentage of time Alice beats Bob to the goal (which is her expected reward for the competitive $r_{\text{joint}}$). On the top row, we plot Bob's expected time steps per unit reward, relative to Alice's. Their combined steps per unit reward would be more directly related to the cooperative $r_{\text{joint}}$ described above, but we plot Bob's individual contribution (relative to Alice's), since his individual contribution to the joint reward rate varies dramatically with $\beta$, whereas Alice's does not. We note that one advantage of our approach is that it unifies cooperative and competitive strategies in the same one-parameter ($\beta$) family.

Below, we will consider two different information regularizers meant to encourage/discourage Alice from sharing goal information with Bob: the (conditional) mutual information between goal and action given state, $I_{\text{action}}[\pi] \equiv I(A; G \mid S)$, which we will call the "action information", and the mutual information between state and goal, $I_{\text{state}}[\pi] \equiv I(S; G)$, which we will call the "state information." Since the mutual information is a general measure of dependence (linear and non-linear) between two variables, $I_{\text{action}}$ and $I_{\text{state}}$ measure the ease in inferring the goal from the actions and states, respectively, generated by the policy $\pi$. Thus, if Alice wants Bob to do well, she should choose a policy with high information, and vice versa if not.

We consider both action and state informations because they have different advantages and disadvantages. Using action information assumes that Bob (the observer) can see both Alice's states *and* actions, which may be unrealistic in some environments, such as one in which the actions are the torques a robot applies to its joint angles [Eysenbach et al., 2019]. Using state information instead only assumes that Bob can observe Alice's states (and not actions), however it does so at the cost of requiring Alice to count goal-dependent state frequencies under the current policy. Optimizing action information, on the other hand, does not require state counting. So, in summary, action information is simpler to optimize, but state information may be more appropriate to use in a setting where an observer can't observe (or infer) the observee's actions.

The generality with which mutual information measures dependence is at once its biggest strength and weakness. On the one hand, using information allows Alice to prepare for interaction with Bob with neither a model of nor interaction with him. On the other hand, Bob might have limited computational resources (for example, perhaps his policy is linear with respect to his observations of Alice) and so he may not be able to "decode" all of the goal information that Alice makes available to him. Nevertheless, $I_{\text{action}}$ and $I_{\text{state}}$ can at least be considered upper bounds on Bob's inference performance; if $I_{\text{action}} = 0$ or $I_{\text{state}} = 0$, it would be impossible for Bob to guess the goal (above chance) from Alice's actions or states, respectively, alone.

Optimizing information can be equivalent to optimizing reward under certain conditions, such as in the following example. Consider Bob's subtask of identifying the correct goal in a 2-goal setup. If his belief over the goal is represented by $p(g)$, then he should guess $g^* = \text{argmax}_g p(g)$, which results in error probability $p_{\text{err}} = 1 - \max_g p(g)$. Since the binary entropy function $H(g) \equiv H[p(g)]$ increases monotonically with $p_{\text{err}}$, optimizing one is equivalent to optimizing the other. Denoting the parts of Alice's behavior observable by Bob as $x$, then $H(g \mid x)$ is the post-observation entropy in Bob's beliefs, and optimizing it is equivalent to optimizing $I(g; x) = H(g) - H(g \mid x)$, since the pre-observation entropy $H(g)$ is not dependent on Alice's behavior. If Bob receives reward $r$ when identifying the right goal, and 0 otherwise, then his expected reward is $(1 - p_{\text{err}}) r$. Thus, in this simplified setup, optimizing information is directly related to optimizing reward. In general, when one considers the temporal dynamics of an episode, more than two goals, or more complicated reward structures, the relationship becomes more complicated. However, information is useful in abstracting away that complexity, and preparing Alice generically for a plethora of possible task setups.

## 2.1 Optimizing action information: $I_{\text{action}} \equiv I(A; G \mid S)$

First, we discuss regularization via optimizing the mutual information between goal and action (conditioned on state), $I_{\text{action}} \equiv I(A; G \mid S)$, where $G$ is the goal for the episode, $A$ is the chosen action, and $S$ is the state of the agent. That is, we will train an agent to maximize the objective $J_{\text{action}}[\pi] \equiv \mathbb{E}[r] + \beta I_{\text{action}}$, where $\beta$ is a tradeoff parameters whose sign determines whether we want the agent to signal (positive) or hide (negative) their intentions, and whose magnitude determines the relative preference for rewards and intention signaling/hiding.

$I_{\text{action}}$ is a functional of the multi-goal policy $\pi_g(a \mid s) \equiv p(a \mid s, g)$, that is the probability distribution over actions given the current goal and state, and is given by:

$$I_{\text{action}} \equiv I(A; G \mid S) = \sum_s p(s) \, I(A; G \mid S = s) \tag{1}$$

$$= \sum_g \rho_G(g) \sum_s p(s \mid g) \sum_a \pi_g(a \mid s) \log \frac{\pi_g(a \mid s)}{p(a \mid s)}. \tag{2}$$

The quantity involving the sum over actions is a KL divergence between two distributions: the goal-dependent policy $\pi_g(a \mid s)$ and a goal-independent policy $p(a \mid s)$. This goal-independent policy comes from marginalizing out the goal, that is $p(a \mid s) = \sum_g \rho_G(g) \, \pi_g(a \mid s)$, and can be thought of as a fictitious policy that represents the agent's "habit" in the absence of knowing the goal. We will denote $\pi_0(a \mid s) \equiv p(a \mid s)$ and refer to it as the "base policy," whereas we will refer to $\pi_g(a \mid s)$ as simply the "policy." Thus, we can rewrite the information above as:

$$I_{\text{action}} = \sum_g \rho_G(g) \sum_s p(s \mid g) \, \text{KL}[\pi_g(a \mid s) \mid \pi_0(a \mid s)] = \mathbb{E}_\tau [\text{KL}[\pi_g(a \mid s) \mid \pi_0(a \mid s)]]. \tag{3}$$

Writing the information this way suggests a method for stochastically estimating it. First, we sample a goal $g$ from $p(g)$, that is we initialize an episode of some task. Next, we sample states $s$ from $p(s \mid g)$, that is we generate state trajectories using our policy $\pi_g(a \mid s)$. At each step, we measure the KL between the policy and the base policy. Averaging this quantity over episodes and steps give us our estimate of $I_{\text{action}}$.

*Optimizing* $I_{\text{action}}$ with respect to the policy parameters $\theta$ is a bit trickier, however, because the expectation above is with respect to a distribution that depends on $\theta$. Thus, the gradient of $I_{\text{action}}$ with

---
**Algorithm 1** Action information regularized REINFORCE with value baseline.
---
    **Input**: $\beta$, $\rho_G$, $\gamma$, and ability to sample MDP $\mathcal{M}$
    Initialize $\pi$, parameterized by $\theta$
    Initialize $V$, parameterized by $\phi$
    **for** $i = 1$ **to** $N_{\text{episodes}}$ **do**
        Generate trajectory $\tau = (g, s_0, a_0, s_1, a_1, \ldots, s_T)$
        **for** $t = 0$ **to** $T - 1$ **do**
            Update policy in direction of $\nabla_\theta J_{\text{action}}(t)$ using equation 6
            Update value in direction of $-\nabla_\phi \left( V_g(s_t) - \tilde{R}_t \right)^2$ with $\tilde{r}(t)$ according to equation 7
        **end for**
    **end for**
---

respect to $\theta$ has two terms:

$$\nabla_\theta I_{\text{action}} = \sum_g \rho_G(g) \sum_s \left( \nabla_\theta p(s \mid g) \right) \mathrm{KL}[\pi_g(a \mid s) \mid \pi_0(a \mid s)] \tag{4}$$

$$+ \sum_g \rho_G(g) \sum_s p(s \mid g) \nabla_\theta \mathrm{KL}[\pi_g(a \mid s) \mid \pi_0(a \mid s)]. \tag{5}$$

The second term involves the same sum over goals and states as in equation 3, so it can be written as an expectation over trajectories, $\mathbb{E}_\tau[\nabla_\theta \mathrm{KL}[\pi_g(a \mid s) \mid \pi_0(a \mid s)]]$, and therefore is straightforward to estimate from samples. The first term is more cumbersome, however, since it requires us to model (the policy dependence of) the goal-dependent state probabilities, which in principle involves knowing the dynamics of the environment. Perhaps surprisingly, however, the gradient can still be estimated purely from sampled trajectories, by employing the so-called "log derivative" trick to rewrite the term as an expectation over trajectories. The calculation is identical to the proof of the policy gradient theorem [Sutton et al., 2000], except with reward replaced by the KL divergence above.

The resulting Monte Carlo policy gradient (MCPG) update is:

$$\nabla_\theta J_{\text{action}}(t) = A_{\text{action}}(t) \nabla_\theta \log \pi_g(a_t \mid s_t) + \beta \nabla_\theta \mathrm{KL}[\pi_g(a \mid s_t) \mid \pi_0(a \mid s_t)], \tag{6}$$

where $A_{\text{action}}(t) \equiv \tilde{R}_t - V_g(s_t)$ is a modified advantage, $V_g(s_t)$ is a goal-state value function regressed toward $\tilde{R}_t$, $\tilde{R}_t = \sum_{t'=t}^{T} \gamma^{t'-t} \tilde{r}_{t'}$ is a modified return, and the following is the modified reward feeding into that return:

$$\tilde{r}_t \equiv r_t + \beta \mathrm{KL}[\pi_g(a \mid s_t) \mid \pi_0(a \mid s_t)]. \tag{7}$$

The second term in equation 6 encourages the agent to alter the policy to share or hide information in the *present* state. The first term, on the other hand, encourages modifications which lead the agent to states in the *future* which result in reward and the sharing or hiding of information. Together, this optimizes $J_{\text{action}}$. This algorithm is summarized in algorithm 2.1.

## 2.2 Optimizing state information: $I_{\text{state}} \equiv I(S; G)$

We now consider how to regularize an agent by the information one's *states* give away about the goal, using the mutual information between state goal, $I_{\text{state}} \equiv I(S; G)$. This can be written:

$$I_{\text{state}} = \sum_g \rho_G(g) \sum_s p(s \mid g) \log \frac{p(s \mid g)}{p(s)} = \mathbb{E}_\tau \left[ \log \frac{p(s \mid g)}{p(s)} \right]. \tag{8}$$

In order to *estimate* this quantity, we could track and plug into the above equation the empirical state frequencies $p_{\text{emp}}(s \mid g) \equiv \frac{N_g(s)}{N_g}$ and $p_{\text{emp}}(s) \equiv \frac{N(s)}{N}$, where $N_g(s)$ is the number of times state $s$ was visited during episodes with goal $g$, $N_g \equiv \sum_s N_g(s)$ is the total number of steps taken under goal $g$, $N(s) \equiv \sum_g N_g(s)$ is the number of times state $s$ was visited across all goals, and $N \equiv \sum_{g,s} N_g(s) = \sum_g N_g = \sum_s N(s)$ is the total number of state visits across all goals and states. Thus, keeping a moving average of $\log \frac{p_{\text{emp}}(s_t \mid g)}{p_{\text{emp}}(s_t)}$ across episodes and steps yields an estimate of $I_{\text{state}}$.

---
**Algorithm 2** State information regularized REINFORCE with value baseline.
---
    **Input**: $\beta$, $\rho_G$, $\gamma$, and ability to sample MDP $\mathcal{M}$
    Initialize $\pi$, parameterized by $\theta$
    Initialize $V$, parameterized by $\phi$
    Initialize the state counts $N_g(s)$
    **for** $i = 1$ **to** $N_{\text{episodes}}$ **do**
        Generate trajectory $\tau = (g, s_0, a_0, s_1, a_1, \ldots, s_T)$
        Update $N_g(s)$ (and therefore $p_{\text{emp}}(s \mid g)$) according to $\tau$
        **for** $t = 0$ **to** $T - 1$ **do**
            Update policy in direction of $\nabla_\theta J_{\text{state}}(t)$ using equation 11
            Update value in direction of $-\nabla_\phi \left( V_g(s_t) - \tilde{R}_t \right)^2$ with $\tilde{r}(t)$ according to equation 12
        **end for**
    **end for**
---

However, we are of course interested in *optimizing* $I_{\text{state}}$ and so, as in the last section, we need to employ a slightly more sophisticated estimate procedure. Taking the gradient of $I_{\text{state}}$ with respect to the policy parameters $\theta$, we get:

$$\nabla_\theta I_{\text{state}} = \sum_g \rho_G(g) \sum_s \left( \nabla_\theta p(s \mid g) \right) \log \frac{p(s \mid g)}{p(s)} \tag{9}$$

$$+ \sum_g \rho_G(g) \sum_s p(s \mid g) \left( \frac{\nabla_\theta p(s \mid g)}{p(s \mid g)} - \frac{\nabla_\theta p(s)}{p(s)} \right). \tag{10}$$

The calculation is similar to that for evaluating $\nabla_\theta I_{\text{action}}$ and details can be found in section S1. The resulting MCPG update is:

$$\nabla_\theta J_{\text{state}}(t) = A_{\text{state}}(t) \, \nabla_\theta \log \pi_g(a_t \mid s_t) - \beta \sum_{g' \neq g} \rho_G\left(g'\right) R_{\text{cf}}\left(t, g, g'\right) \nabla_\theta \log \pi_{g'}(a_t \mid s_t), \tag{11}$$

where $A_{\text{state}}(t) \equiv \tilde{R}_t - V_g(s_t)$ is a modified advantage, $V_g(s_t)$ is a goal-state value function regressed toward $\tilde{R}_t$, $\tilde{R}_t \equiv \sum_{t'=t}^{T} \gamma^{t'-t} \tilde{r}_{t'}$ is a modified return, $R_{\text{cf}}\left(t, g, g'\right) \equiv \sum_{t'=t}^{T} \gamma^{t'-t} r_{\text{cf}}\left(t', g, g'\right)$ is a "counterfactual goal return", and the following are a modified reward and a "counterfactual goal reward", respectively, which feed into the above returns:

$$\tilde{r}_t \equiv r_t + \beta \left( 1 - p_{\text{emp}}(g \mid s_t) + \log \frac{p_{\text{emp}}(s_t \mid g)}{p_{\text{emp}}(s_t)} \right) \tag{12}$$

$$r_{\text{cf}}\left(t, g, g'\right) \equiv \left( \prod_{t'=0}^{t} \frac{\pi_{g'}(a_{t'} \mid s_{t'})}{\pi_g(a_{t'} \mid s_{t'})} \right) \frac{p_{\text{emp}}(s_t \mid g)}{p_{\text{emp}}(s_t)}, \tag{13}$$

where $p_{\text{emp}}(g \mid s_t) \equiv \rho_G(g) \frac{p_{\text{emp}}(s_t \mid g)}{p_{\text{emp}}(s_t)}$. The modified reward can be viewed as adding a "state uniqueness bonus" $\log \frac{p_{\text{emp}}(s_t \mid g)}{p_{\text{emp}}(s_t)}$ that tries to increase the frequency of the present state under the present goal to the extent that the present state is more common under the present goal. If the present state is less common than average under the present goal, then this bonus becomes a penalty. The counterfactual goal reward, on the other hand, tries to make the present state less common under *other* goals, and is again scaled by uniqueness under the present goal $\frac{p_{\text{emp}}(s_t \mid g)}{p_{\text{emp}}(s_t)}$. It also includes importance sampling weights to account for the fact that the trajectory was generated under the current goal, but the policy is being modified under other goals. This algorithm is summarized in algorithm 2.2.

## 3 Related work

Whye Teh et al. [2017] recently proposed an algorithm similar to our action information regularized approach (algorithm 2.1), but with very different motivations. They argued that constraining goal-specific policies to be close to a distilled base policy promotes transfer by sharing knowledge

across goals. Due to this difference in motivation, they only explored the $\beta < 0$ regime (i.e. our "competitive" regime). They also did not derive their update from an information-theoretic cost function, but instead proposed the update directly. Because of this, their approach differs in that it did not include the $\beta \nabla_\theta \text{KL}[\pi_g \mid \pi_0]$ term, and instead only included the modified return. Moreover, they did not calculate the full KLs in the modified return, but instead estimated them from single samples (e.g. $\text{KL}[\pi_g(a \mid s_t) \mid \pi_0(a \mid s_t)] \approx \log \frac{\pi_g(a_t \mid s_t)}{\pi_0(a_t \mid s_t)}$). Nevertheless, the similarity in our approaches suggest a link between transfer and competitive strategies, although we do not explore this here.

Eysenbach et al. [2019] also recently proposed an algorithm similar to ours, which used both $I_{\text{state}}$ and $I_{\text{action}}$ but with the "goal" replaced by a randomly sampled "skill" label in an unsupervised setting (i.e. no reward). Their motivation was to learn a diversity of skills that would later would be useful for a supervised (i.e. reward-yielding) task. Their approach to optimizing $I_{\text{state}}$ differs from ours in that it uses a discriminator, a powerful approach but one that, in our setting, would imply a more specific model of the observer which we wanted to avoid.

Tsitsiklis and Xu [2018] derive an inverse tradeoff between an agent's delay in reaching a goal and the ability of an adversary to predict that goal. Their approach relies on a number of assumptions about the environment (e.g. agent's only source of reward is reaching the goal, opponent only need identify the correct goal and not reach it as well, nearly uniform goal distribution), but is suggestive of the general tradeoff. It is an interesting open question as to under what conditions our information-regularized approach achieves the optimal tradeoff.

Dragan et al. [2013] considered training agents to reveal their goals (in the setting of a robot grasping task), but did so by building an explicit model of the observer. Ho et al. [2016] uses a similar model to capture human generated actions that "show" a goal also using an explicit model of the observer. There is also a long history of work on training RL agents to cooperate and compete through interactive training and a joint reward (e.g. [Littman, 1994, 2001, Kleiman-Weiner et al., 2016, Leibo et al., 2017, Peysakhovich and Lerer, 2018, Hughes et al., 2018]), or through modeling one's effect on another agent's learning or behavior (e.g. [Foerster et al., 2018, Jaques et al., 2018]). Our approach differs in that it requires neither access to an opponent's rewards, nor even interaction with or a model of the opponent. Without this knowledge, one can still be cooperative (competitive) with others by being as (un)clear as possible about one's own intentions. Our work achieves this by directly optimizing information shared.

# 4 Experiments

We demonstrate the effectiveness of our approach in two stages. First, we show that training Alice (who has access to the goal of the episode) with information regularization effectively encourages both goal signaling and hiding, depending on the sign of the coefficient $\beta$. Second, we show that Alice's goal signaling and hiding translate to higher and lower rates of reward acquisition for Bob (who does not have access to the goal and must infer it from observing Alice), respectively. We demonstrate these results in two different simple settings. Our code is available at https://github.com/djstrouse/InfoMARL.

## 4.1 Spatial navigation

The first setting we consider is a simple grid world spatial navigation task, where we can fully visualize and understand Alice's regularized policies. The $5 \times 5$ environment contains two possible goals: the top left state or the top right. On any given episode, one goal is chosen randomly (so $\rho_G$ is uniform) and that goal state is worth $+1$ reward. The other goal state is then worth $-1$. Both are terminal. Each of Alice and Bob spawn in a random (non-terminal) state and take actions in $\mathcal{A} = \{\text{left}, \text{right}, \text{up}, \text{down}, \text{stay}\}$. A step into a wall is equivalent to the stay action but results in a penalty of $-.1$ reward. We first train Alice alone, and then freeze her parameters and introduce Bob.

Alice was trained using implementations of algorithms 2.1 and 2.2 in TensorFlow [Abadi et al., 2016]. Given the small, discrete environment, we used tabular representations for both $\pi$ and $V$. See section S2.1 for training parameters.

Examples of Alice's resulting policies are shown in figure 1. The top row contains policies regularized with $I_{\text{action}}$, the bottom with $I_{\text{state}}$. The left column contains "cooperative" policies encouraged to share goal information ($\beta = .025$), the middle "ambivalent" policies that are unregularized ($\beta = 0$),

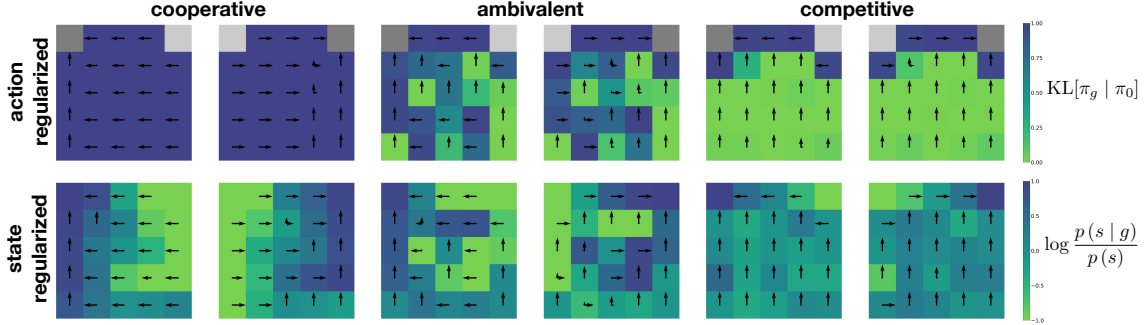

Figure 1: **Information-regularized policies**. Top row: regularization with $I_{\text{action}}$. Bottom row: regularization with $I_{\text{state}}$. Left column: $\beta = .025$. Center column: $\beta = 0$. Right column: $\beta = .025$. See main text for additional details.

and the right "competitive" policies encouraged to hide goal information ($\beta = -.025$). In each case, two policies are shown - at left, the policy when the goal is in the top left, and at right, when it is in the top right. The arrows are proportional to Alice's action probabilities. The color scale indicates $\text{KL}[\pi_g \mid \pi_0]$ and $\log \frac{p_{\text{emp}}(s|g)}{p_{\text{emp}}(s)}$ in bits for action and state information regularization, respectively. Thus, bluer indicates more goal information shared in that state.

In the cooperative case with action information regularization, Alice wants to maximize $\text{KL}[\pi_g \mid \pi_0]$ and thus she wants her goal-dependent policies to differ as much as possible. This primarily results in her going left in most states when the goal is in the top left, and right when its in the top right. She can also choose to have some probability of going up in those states, as long as she only does so under one goal or the other and not both. Ultimately this means that no matter what state Alice spawns in, she immediately signals the goal with her very first action, and continues to do so with each subsequent action.

In the competitive case with action information regularization, Alice instead wants to *minimize* $\text{KL}[\pi_g \mid \pi_0]$ and thus she wants her goal-dependent policies to *match* as much as possible. In the present case, this means that she chooses to go up as far as she can until she must pivot in order to still reach the goal in the minimal number of steps. This leads to her hiding the goal for as long as possible without sacrificing reward.

Without information regularization, Alice's policy is a random mixture of the competitive and cooperative strategies, the details of which are determined by initialization and the randomness of training trajectories. Regularization with state information yields similar policies to those generated with action information in all three cases, since in this deterministic setting, different action choices are tightly coupled with different state trajectories.

To demonstrate that Alice's goal revealing and hiding behaviors are useful for cooperation and competition, respectively, we then trained a second agent, Bob, who does not have access to the goal and instead must infer it from observing Alice. Thus, while Alice's inputs at time $t$ were the present goal $g$ and her state $s_t^{\text{alice}}$, Bob's are Alice's present state and action $s_t^{\text{alice}}$ and $a_t^{\text{alice}}$, as well as his own state $s_t^{\text{bob}}$. Details are available in section S2.1, but in brief, Bob processes Alice's state-action trajectories with an RNN to form a belief about the goal, which then feeds into his policy, all of which is trained end-to-end via REINFORCE.

We trained 5 of each of the 3 versions of Alice above, and 10 Bobs per Alice. We plot the results for the best performing Bob for each Alice (so $5 \times 3 = 15$ curves) in figure 2. We use all 5 Alices to estimate the variance in our approach, but the best-of-10 Bob to provide a reasonable estimate of the *best* performance of a friend/foe.

We measure Bob's performance in terms of his episode length, relative to Alice's, as well as the percentage of time he beats Alice to the goal. For both action and state information regularization, encouraging Alice to hide goal information leads to Bob taking about 30% longer to reach the goal relative to when Alice is encouraged to share goal information. Information-hiding Alice receives a boost of similar magnitude in the frequency with which she beats Bob to the goal. Training without information regularization leads to results in between the competitive and cooperative strategies,

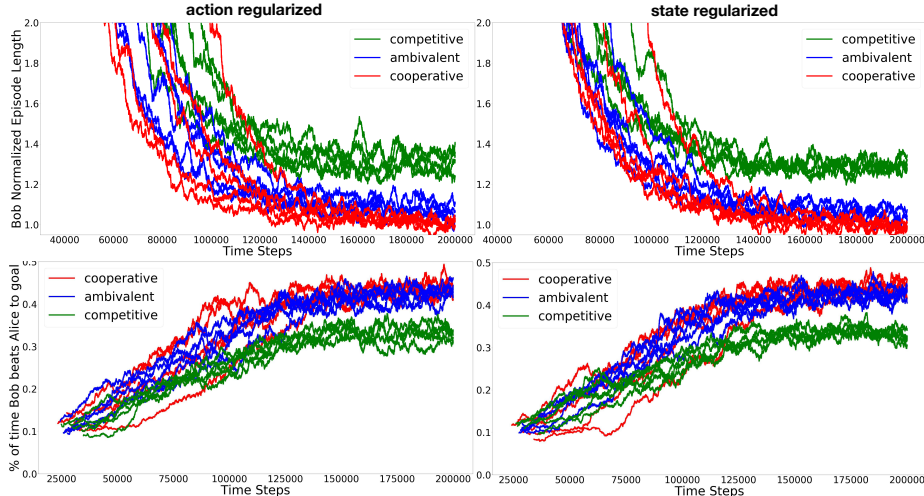

Figure 2: **The effect of Alice hiding/sharing goal information on Bob's performance.** Left column: regularization with $I_{\text{action}}$. Right column: regularization with $I_{\text{state}}$. Top row: Bob's episode length relative to Alice's (moving average over 500 episodes). Bottom row: the percentage of time Alice beats Bob to the goal (moving average over 1000 episodes).

although closer to the cooperative strategy in this case. We also note that the variance in Bob's performance was higher for the unregularized case, and much higher for the competitive case, with nearly zero variance in performance for the cooperative case, indicating that information hiding and sharing make training harder and easier, respectively.

### 4.2 Key-and-door game

In the above spatial navigation task, information regularization of Alice breaks symmetries between equally-preferred (in terms of reward) navigation strategies. However, in many scenarios, it might be worthwhile for an agent to give up some reward if it means large gains in the ability to hide or share information. To demonstrate that our approach could also discover such "lossy" strategies, we designed a simple key-and-door game with this feature (figure 3, left).

It is again a two-goal (door) game with the same action space and reward structure as the spatial navigation setting. Alice again alone receives the goal, and Bob must infer it from observing her. The difference is that, in order to enter the terminal states, Alice and Bob must first pick up an appropriate key. Each agent has goal-specific keys that only they can pick up (top/bottom rows, color-coded to door, labeled with A/B for Alice/Bob). Alice also has access to a master key that can open both doors (center right). Agents can only pick up one key per episode - the first they encounter. Bob spawns in the same location every time (the "B"), while Alice spawns in any of the 3 spaces between her two goal-specific keys (the "A" and spaces above/below). This means that Bob has a shorter path to the goals, and thus if Alice telegraphs the goal right away, Bob will beat her to it. While Alice's master key is strictly on a longer path to the goal, picking it up allows her to delay informing Bob of the goal such that she can beat him to it.

We trained Alice with action information regularization as in the previous section (see section S2.2 for training parameters). When unregularized or encouraged to share goal information ($\beta = .25$), Alice simply took the shortest path to the goal, never picking up the master key. When Bob was trained on these Alices, he beat/tied her to the goal on approximately 100% of episodes (figure 3, right). When encouraged to hide information ($\beta = -.25$), however, we found that Alice learned to take the longer path via the master key on about half of initializations (example in figure 3, center). When Bob was trained on these Alices, he beat/tied her to the goal much less than half the time (figure 3, right). Thus, our approach successfully encourages Alice us to forgo rewards during solo training in order to later compete more effectively in an interactive setting.

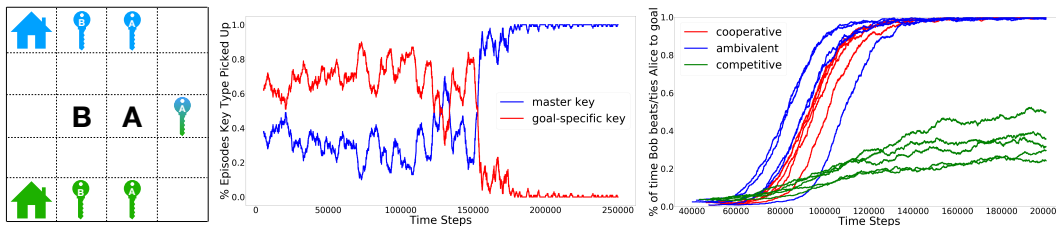

Figure 3: **Key-and-door game results**. Left: depiction of game. Center: percentage episodes in which Alice picks up goal-specific vs master key during training in an example run (moving average over 100 episodes). Right: percentage episodes in which Bob beats/tie Alice to the goal (moving average over 1000 episodes).

## 5 Discussion

In this work, we developed a new framework for building agents that balance reward-seeking with information-hiding/sharing behavior. We demonstrate that our approach allows agents to learn effective cooperative and competitive strategies in asymmetric information games without an explicit model or interaction with the other agent(s). Such an approach could be particularly useful in settings where interactive training with other agents could be dangerous or costly, such as the training of expensive robots or the deployment of financial trading strategies.

We have here focused on simple environments with discrete and finite states, goals, and actions, and so we briefly describe how to generalize our approach to more complex environments. When optimizing $I_{\text{action}}$ with many or continuous actions, one could stochastically approximate the action sum in $\text{KL}[\pi_g \mid \pi_0]$ and its gradient (as in [Whye Teh et al., 2017]). Alternatively, one could choose a form for the policy $\pi_g$ and base policy $\pi_0$ such that the KL is analytic. For example, it is common for $\pi_g$ to be Gaussian when actions are continuous. If one also chooses to use a Gaussian approximation for $\pi_0$ (forming a variational bound on $I_{\text{action}}$), then $\text{KL}[\pi_g \mid \pi_0]$ is closed form. For optimizing $I_{\text{state}}$ with continuous states, one can no longer count states exactly, so these counts could be replaced with, for example, a pseudo-count based on an approximate density model. [Bellemare et al., 2016, Ostrovski et al., 2017] Of course, for both types of information regularization, continuous states or actions also necessitate using function approximation for the policy representation. Finally, although we have assumed access to the goal distribution $\rho_G$, one could also approximate it from experience.

## Acknowledgements

The authors would like to acknowledge Dan Roberts and our anonymous reviewers for careful comments on the original draft; Jane Wang, David Pfau, and Neil Rabinowitz for discussions on the original idea; and funding from the Hertz Foundation (DJ and Max), The Center for Brain, Minds and Machines (NSF #1231216) (Max and Josh), the NSF Center for the Physics of Biological Function (PHY-1734030) (David), and as a Simons Investigator in the MMLS (David).

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
