[Supplementary Material]

# Supplemental Materials

## S1   Calculating $\nabla_\theta I_{\mathbf{state}}(t)$

We want to evaluate:

$$\nabla_\theta I_{\mathrm{state}} = \sum_g \rho_G(g) \sum_s (\nabla_\theta p(s \mid g)) \log \frac{p(s \mid g)}{p(s)} \tag{S1}$$

$$+ \sum_g \rho_G(g) \sum_s p(s \mid g) \frac{\nabla_\theta p(s \mid g)}{p(s \mid g)} \tag{S2}$$

$$- \sum_g \rho_G(g) \sum_s p(s \mid g) \frac{\nabla_\theta p(s)}{p(s)} \tag{S3}$$

$$\equiv T_1 + T_2 - T_3, \tag{S4}$$

where we denote the three terms by $T_1$, $T_2$, and $T_3$. The effect of $T_1$ follows from the policy gradient theorem and amounts to adding the following to the reward return:

$$\sum_{t'=t}^{T} \log \frac{p_{\mathrm{emp}}(s_{t'} \mid g)}{p_{\mathrm{emp}}(s_{t'})}. \tag{S5}$$

By the same argument, $T_2 = \sum_g p(g) \sum_s \nabla_\theta p(s \mid g)$ simply results in the addition of 1 to the info return at each time step.

Finally, we have the third term:

$$T_3 = \sum_g \rho_G(g) \sum_{s_t} \frac{p(s_t \mid g)}{p(s_t)} \nabla_\theta p(s_t) \tag{S6}$$

$$= \sum_g \rho_G(g) \sum_{s_t} \frac{p(s_t \mid g)}{p(s_t)} \nabla_\theta \sum_{g'} \rho_G(g') \rho_S(s_0) \prod_{t'=0}^{t} \pi_{g'}(a_{t'} \mid s_{t'}) P(s_{t'+1} \mid s_{t'}, a_{t'}) \tag{S7}$$

$$= \sum_g \rho_G(g) \sum_{s_t} \frac{p(s_t \mid g)}{p(s_t)} \sum_{g'} \rho_G(g') \rho_S(s_0) \prod_{t'=0}^{t} \left(\nabla_\theta \pi_{g'}(a_{t'} \mid s_{t'})\right) P(s_{t'+1} \mid s_{t'}, a_{t'}) \tag{S8}$$

$$= \sum_{g,s_t} \rho_G(g) \, \rho_S(s_0) \prod_{t'=0}^{t} \pi_g(a_{t'} \mid s_{t'}) P(s_{t'+1} \mid s_{t'}, a_{t'}) \times \tag{S9}$$

$$\sum_{g'} \rho_G(g') \frac{\pi_{g'}(a_{t'} \mid s_{t'})}{\pi_g(a_{t'} \mid s_{t'})} \frac{\nabla_\theta \pi_{g'}(a_{t'} \mid s_{t'})}{\pi_{g'}(a_{t'} \mid s_{t'})} \frac{p(s_t \mid g)}{p(s_t)} \tag{S10}$$

$$= \mathbb{E}_\tau \left[ \sum_{g'} \rho_G(g') \prod_{t'=0}^{t} \frac{\pi_{g'}(a_{t'} \mid s_{t'})}{\pi_g(a_{t'} \mid s_{t'})} \left(\nabla_\theta \log \pi_{g'}(a_{t'} \mid s_{t'})\right) \frac{p(s_t \mid g)}{p(s_t)} \right] \tag{S11}$$

where in the fourth line we multiply and divide by the policy under both $g$ and $g'$ in order to employ the log derivative trick and to express the equation as an expectation under the present goal. The end result is the update in equation 11.

## S2   Experimental parameters and details

### S2.1   Simple spatial navigation

In order to allow Bob to integrate information about the goal over time and remember it to guide future actions, we endow Bob with a recurrent neural network (RNN) to process Alice's state-action

pairs. We used a gated recurrent unit (GRU) Cho et al. [2014] to which Alice's state-action pairs are fed as a one-hot vector. We chose to use a scalar core state for the GRU since it was simply tasked with tracking Bob's belief about one of two goals, and could thus assign each goal to a sign of the GRU core state/output, which is what Bob chose to do in practice. The GRU output $z_t = \text{RNN}\big(s_t^{\text{alice}}, a_t^{\text{alice}}\big)$ was then concatenated with a one-hot representation of Bob's own state $s_t^{\text{bob}}$ and fed into a fully-connected, feed-forward layer of 128 units with two readout heads: a policy head (a linear layer with $|\mathcal{A}|$ units followed by a softmax, yielding $a_t^{\text{bob}} \sim \pi^{\text{bob}}\big(s_t^{\text{bob}}, z_t\big)$) and a value head (a single linear readout node, yielding $v_t = V^{\text{bob}}\big(s_t^{\text{bob}}, z_t\big)$).

| | Alice | Bob |
|---|---|---|
| training time, in steps | 100k | 200k |
| max episode length, in steps | 100 | 100 |
| entropy bonus (logarithmically annealed from/to) | .5, .005 | .5, .01 |
| learning rate (Adam) | $2.5 \times 10^{-2}$ | $5 \times 10^{-5}$ |
| weight on value function regression term | .5 | .5 |
| discount $\gamma$ | .8 | .8 |

Table 1: **Training parameters**.

## S2.2 Key game

The only difference from the previous set of training parameters is that Alice now trains longer (250k instead of 100k steps).

| | Alice | Bob |
|---|---|---|
| training time, in steps | 250k | 200k |
| max episode length, in steps | 100 | 100 |
| entropy bonus (logarithmically annealed from/to) | .5, .005 | .5, .01 |
| learning rate (Adam) | $2.5 \times 10^{-2}$ | $5 \times 10^{-5}$ |
| weight on value function regression term | .5 | .5 |
| discount $\gamma$ | .8 | .8 |

Table 2: **Training parameters**.