[Reviews · NeurIPS 2018]

Reviewer 1



This paper describes a method for regularizing an agent’s policy to either advertise or hide the agent’s goal from an observer using information theoretic regularization. They consider an MDP where a goal is revealed only to one agent, Alice, but not to an observer, Bob. If Alice is told Bob is a friend (foe) her policy is regularized to maximize (minimize) the mutual information between the goal and either the actions she takes or the state. Significance: I’ll start by evaluating this paper’s significance because I think this is where it’s biggest issue lies. To me, this paper reads as if the authors have a solution without being quite clear on what the problem is. A loss function with mutual information regularization is presented the solution but we are never formally told what the problem is. We are only told that Alice wants to make Bob’s task of inferring her goal either easier or harder depending on whether Bob is a friend or foe. Information regularization sounds like it might be the right solution but I would have liked to see a formal derivation. The reason I think this is important (and not just unnecessary formalism) is I’m not convinced that this solution solves that problem laid out in the introduction. The stated high-level goal was to train an agent such that the observer can infer its intentions from its actions (so that human operators can understand them). But this solution requires supplying the agent with a goal (because it forms part of the regularizer) while not revealing the goal to the observer. I don’t see how this aligns with the motivation that the observer is a system designer who doesn’t understand the agent’s intentions? Why wouldn’t the system designer know the agent’s objective function? The technique seems more useful in the competitive setting where the agent is trying to hide information (there the observer is the opponent), but I would have liked to see a motivating game (in the game theoretic sense) for which information regularization is a good (optimal?) strategy. If we accept that this is a solution to a useful problem the paper is well written and seems technically solid (albeit with an empirical evaluation that focuses only on toy problems). But I’m not convinced it is, so I think in it’s current form it is a good workshop paper that needs a clear motivating example to be turned into a NIPS paper. Quality: - The paper derives the appropriate policy gradient updates that are required in order to apply mutual information regularization. The derivation is relatively lengthly, but as far as I can tell follows standard score function estimator-style arguments. - The experimental evaluation tests the effect of adding information regularization two simple grid world problems. The spatial navigation task gives clear intuition for how the regularization breaks ties in an under constrained problem while the key-door problem shows the trade-offs between information regularization and optimizing the efficient direct policy. - I liked the toy problems for giving intuition about how the method works in very simple domains, but I’m not sure what the experiments are testing beyond “is our Monte Carlo policy gradient update correct?”. I would have liked to see some evaluation that answers “is this the right approach to the problem?” against a strong baseline. Again, it’s tricky to suggest what this experiment would be without a clear motivating example (these toy problems don’t make to the system-designer / observer problems laid out in the intro). - These are very small toy problems and there was no mention of how the method scales to more real-world problems in higher dimensions. Clarity: The paper is well-written and clear. Originality: I’m not aware of previous work with exactly this approach, information theoretic regularization has been used in [Florensa, Duan, Abbeel 2017] (which should have been cited). Typos: Line 69: citation should be before the fullstop Line 70: “relaxes the assumption *to* that Bob…” remove to Line 82: “Alice’s actions or *states*..”

Reviewer 2



This article builds on previous work on signalling in multi-agent games, and incorporates mutual information in trajectories into a regularisation term to learn policies that do or do not reveal information in cooperative and competitive domains respectively. It presents an interesting algorithmic extension but does not convey the motivating significance of learning this new capacity. While the ability to learn either behaviour is clearly beneficial and interesting from a system-designer point of view, it is unclear why an agent would sacrifice reward in the given empirical evaluation. The method resembles reward shaping, but the newly introduced rewards do not diminish, and actually change asymptotically optimal behaviour. It would seem much more interesting to understand whether this method helps in games where the reward actually drops if the opponent infers the goal well enough. In general, the article is well written, and the presentation is rather clear. Related work is cited and the method is contrasted against alternative approaches. Minor comments, on the first page 'whether' should be 'where' and 'composes' might better be 'combines'.

Reviewer 3



Summary: In the framework of a (multi-goal) MDP, one agent (Alice) samples and optimizes a goal, while another agent (Bob) is trying to learn the goal Alice is optimizing from observing states and/or action trajectories. The authors propose two methods, based on mutual information, that allow Alice to learn to reveal (cooperative) or to hide (competitive) her intentions (goal) from Bob while training. Both methods are based on the idea of information regularization, where the objective function of Alice includes a mutual information term; both cooperation and coordination can be handled in the current framework by controlling the sign of the regularization coefficient. However, the methods differ in that the first method, called “action information regularization”, optimizes the mutual information between goal and action given state, whereas the second, called “state information regularization”, optimizes the mutual information between state and goal. The authors show how the two forms of the mutual information can be optimized within a policy gradients (REINFORCE) framework, and propose two algorithms to do so. They illustrate the effectiveness of Alice’s approach in both cooperative and competitive settings across two small examples. Contributions: The main contributions of the paper are as follows: 1. Two methods, state and action information regularization, are proposed to allow an agent to reveal or hide their intentions in the presence of information asymmetry, in the policy gradient setting, and is scalable for deep reinforcement learning 2. By following information regularization, the current method allows the agent to cooperate (signal intentions) or compete (hide intentions) by choosing a positive or negative regularization coefficient Overall comments: I think this is an exceptionally written paper, with a strong motivation and novel theoretical and algorithmic insights. I very much like the information theoretical approach presented here for policy gradients, and how it is able to handle both cooperation and competition, in a very cohesive mathematical framework. In particular, the information penalization method proposed here addresses clearly the tradeoff between optimizing immediate reward and the long-term effect of hiding or sharing current information that can influence the long-term reward. I would like to re-iterate that the ability to handle both cooperation and competition cohesively is a very strong contribution in my view. Motivation: The motivation presented in the current draft is quite strong. However, I do not believe that the authors have argued why their work is relevant for real-world practical examples that would be difficult to solve with existing methods, but have instead only provided abstract and general arguments. Empirical Evaluation: The numerical examples considered are simple but were well-chosen to illustrate the main features in the current framework and lead to some insightful conclusions. In particular, the second domain clearly illustrates the tradeoff that can exist in many problems between a higher immediate or short-term rewards with the long-term rewards that can be gained by hiding or sharing information. Some general comments: 1. It is interesting to see that in both domains, the ambivalent strategy and cooperative strategy perform roughly the same from Bob’s standpoint. It seems plausible to view the ambivalent strategy as a combination of the cooperative and competitive strategy, in that some information regarding the goal is revealed gradually as opposed to in the beginning, as in the cooperative case. My guess is that for larger domains, the difference between the two would become larger, but it is not visible here. Perhaps, if the authors would consider a somewhat “scaled-up” version of the problem(s) (say a 10-by-10 grid-world), the distinction would be clearer. But this is of minor importance. 2. In both examples provided in the current draft, it is clear that Alice is able to hide her intention clearly from Bob and to increase her chance of winning. In the grid-world example, Alice can defer the left or right actions until near the goal. In the key-and-door example, Alice can take the longer route and select the master key. However, there are clearly cases where the opportunities to hide or reveal intentions can be quite limited and ambiguous, due to the structure of the problem. In a noisy grid-world example where actions are no longer deterministically selected, it may be more difficult to reveal intentions with the current approach. In the current grid-world except where the goal states are moved closer and stacked vertically with respect to the initial position, it may not be possible to communicate the intention to Bob until Alice is close to the goal. Such features can make it difficult for Alice to communicate her intention when goals are relatively indistinguishable with respect to KL-divergence, in an information theoretic setting. To me, it is not at all clear how the algorithm will behave and whether or not Bob will be able to respond well to Alice’s attempt to reveal or hide information once both agents are close to the goal. I think it might be important to provide examples or reasoning to highlight and clarify these issues with respect to the current method. Minor comments: On line 221, the reference to Figure 4.1 should probably be changed to 4.2. Similarly, from lines 251-259 the references to Figure 4.2 should likely be 4.3.

Reviewer 4



The authors present a framework, in the context of 2 player stochastic games, that enables an agent to (hide) reveal intentions to a second agent by executing (non-) informative actions or visiting (non-) informative states. The revealed information can be used by the second agent to infer what is the goal of the stochastic game. This can be achieved by extending the usual RL objective in two possible ways. On one hand, one can add a mutual information bonus (or penalty) between states and goals, and on the other hand, between actions and goals given states. The paper presents how to estimate the gradients of those mutual informations in order to apply policy gradient updates. Some experiments in a grid world domain demonstrate the agent’s behaviour when hiding/revealing such information to the second agent. The paper is overall a good paper. The parts of the paper with no math is well written but for the mathy sections it could benefit from a more accurate exposition (defining all terms properly), and in particular from a more formal exposition of the true objective function that algorithm is optimizing (I will elaborate more into that later). In general, though, it is an interesting and well executed idea. It seems an original idea to use the mutual information in this way and clearly one can see the effect on the agent behaviour to hide/reveal intentions. It would have been very nice to extend the results to more complex / or high dimensional problems, since at this current state the applicability is rather limited. A part from this, it is a nice work. I have two main concerns: 1) On the exposition: It is clear that the authors want to maximize the sum of future discounted rewards (and KL terms). However, for example, as noted at the end of page 2, the authors mention “… we will train the agent to maximize the objective J_action[\pi] = E[r] + \beta I_action …” where it seems that the expectation of rewards is computed for a single time-step ( and also the expectation inside I_action). Then later after all the gradients derivation, one reads in Eq 6 that the gradient (now, suddenly depending on time t) depends actually on the sum of rewards (that are inside A_action(t) ). Similarly, the variables (such as the state and actions) in the I_action term that were time-independent according to Eq 1, suddenly they are time-dependent in Eq 6. I think (might be wrong here) that the authors define the problem formulation with a mutual-information for a single-time step, but then really want to optimize a sum of mutual-informations (as one can see from Equation 7). They, somehow, achieve their final goal by doing the n-th order approximations to the densities \hat p_n (s_t |g) and \hat p_n (s_t) and then re-arranging terms in a certain way (for example when going from Eq 21 to Eq 22 in the appendix). In general, I believe that the paper and the derivations would be clearer if the authors would properly define the objective that they are optimizing. 2) In the tables 1 and 2 in the appendix one can see that the n-th order approximation of \hat p_n (s_t |g) and \hat p_n (s_t) is made with n being 0 and 1 for the action and state mutual informations respectively. I expected to see bigger number here. Why did the authors choose the the zeroth order for the action mutual information, since they actually can compute a better approximation? Minor comments: - I_action in line 85 is already defined in line 60 - p(s) in Eq 1 is not properly defined. What distribution over states, the stationary? The time-dependent distribution? - Eq 5 of the appendix, If I am not wrong, there is an s_{t-u} that should be s_{t-n} - Eq 6 of the appendix, If I am not wrong, the sum is up to “n” not up to “u”